# The Supramolecular Organogel Formed by Self-Assembly of Ursolic Acid Appended with Aromatic Rings

**DOI:** 10.3390/ma12040614

**Published:** 2019-02-18

**Authors:** Jinrong Lu, Jinshan Hu, Yinghua Liang, Wenquan Cui

**Affiliations:** College of Chemical Engineering, Hebei Key Laboratory for Environment Photocatalytic and Electrocatalytic Materials, North China University of Science and Technology, Tangshan 063210, China; lujinrong@ncst.edu.cn (J.L.); jinshanhu@ncst.edu.cn (J.H.); liangyh@ncst.edu.cn (Y.L.)

**Keywords:** ursolic acid, derivative, supramolecular organogel

## Abstract

Ursolic acid (UA) as a natural ursane-triterpenoid has rich pharmacological activities. We have found that it possesses aggregation properties and could self-assemble into organogels. Based on the aggregation property of ursolic acid in suitable solvents, its derivative appended with aromatic rings by amide groups was synthesized. The property of self-assembly into organogel was studied in this paper. The results revealed that this derivative could form supramolecular gel in halogenated benzene and also gelate chloroform in the presence of toluene or *p*-xylene. By Fourier-transform infrared spectra (FT-IR) and variable temperature proton nuclear magnetic resonance (^1^H NMR), it was proved that intermolecular hydrogen bonding and π–π stacking interaction were the primary driving forces for the aggregation to form organogel.

## 1. Introduction

Supramolecular gel formed by the self-assembly of someorganic molecules into a three-dimensional network which can trap solvent molecules is an attractive part of supramolecular chemistry [1,2]. Small organic molecules called gelators constitute the architectures of supramolecular gels. Specific functions and properties of gels can be achieved by designing gelators between which the interactions such as hydrogen bonding, π–π stacking interaction, van der Waals forces, charge-transfer and electrostatic interactions, which drive the formation of their architectures [3,4,5]. Due to the ability of bottom-up self-assembly through the above interactions, biomolecules such as steroids, nucleobases and amino acids are attractive candidates to construct supramolecular gels through simple chemical modification [6,7,8,9]. Therefore, imitating and modifying natural products as well-defined building blocks for self-assembly could minimize tedious synthesis work. 

Among natural products, triterpenoids characterized by biocompatibility, rigid chiral skeletons and chemical reaction sites were also found to have strong aggregation trends in suitable solvents [10]. For example, lupine-triterpenoid arjunolic acid [11,12], oleanane-triterpenoid glycyrrhetinic acid [13,14,15,16,17,18], betulinic acid [19,20] and oleanolic acid [21,22] could all aggregate in suitable solvents and form supramolecular gel. Based on their aggregation properties, some functional groups which could provide additional non-covalent interactions could be grafted into triterpenoid skeletons to developsupramolecular self-assembly systems. So, a series of self-assembly structures based on their functional derivatives have been constructed during the last decades [23]. 

Ursolic acid (UA) (Scheme 1) in the form of ursane-triterpenoids has demonstrated anticancer, anti-inflammatory, antioxidant and hypotensivepharmacological activities [24,25,26]. Hence, it is widely used in medicine and natural whitening cosmetics. The aggregation property of unmodified ursolic acid was first reported by the authors in 2016 [27]. It was illustrated that ursolic acid could self-assemble into organogels in bromobenzene and some alcoholic solvents, which proved the aggregation property of the rigid chiral skeleton of ursolic acid. Hypothetically, designing derivatives of ursolic acid conjugated with functional groups which could provide non-covalent interactions may develop functional supramolecular assemblies based on ursolic acid. However, there are few studieson the self-assembly properties of its functional molecules, which are important for their applications. As we know, π–π stacking and hydrogen bonding are the more effective and commonly used non-covalent interactions. Moreover, π–π stacking of aromatic rings can reduce steric repulsion [28,29]. Herein, in order to develop the self-assembly system based on ursolic acid, the derivative **UA-5** (Scheme 1) of ursolic acid appended with aromatic rings was designed. Benzene rings were connected to UA by amide groups which could provide hydrogen bonding interactions. The self-assembly properties were studied, and the results demonstrated that UA couldself-assemble into nanofibers and form a 3D network in a mixture of chloroform and aromatic solvents. It was proved that the driving forces were acombination of hydrogen bonding, π–π stacking and van der Waals forces.

## 2. Materials and Methods 

### 2.1. General Procedure

Chromatographic purifications were performed by column chromatography using 0.06–0.20 mm silica gel. NMR spectra were measured with a JEOL JNM-ECA 300/600 instrument (JEOL Ltd, Tokyo, Japan). A Bruker Esquire-LC spectrometer (Bruker, Fallanden, Switzerland) was employed to measure electrospray ionization mass spectrometry (ESI-MS). TEM was performed using a Tecnai Spiri 120 KV instrument (FEI, Hillsboro, OR, USA). Rheological properties were measured by a Physica MCR301 rotary rheometer (Anton Paar, Germany) at room temperature.

Triethylamine (Et_3_N), benzoyl chloride, toluene, chlorobenzene, *o*-dichlorobenzene, bromobenzeneand, DMF were purchased from Beijing Chemical Plant (analytical purity). All solid samples requiring drying were vacuum-dried for 5–10 h before use.

*p*-phenylenediamine, 1-ethyl-3-(dimethylaminopropyl) carbonyl diimide hydrochloride (EDCI), trifluoroacetic acid (TFA) and isophthaloyl dichloride were purchased from J&K Scientific Co., Ltd. (Beijing, China).

### 2.2. Synthesis

*Compound **UA-1***: Initially, 5.0 g (10.4 mmol) of ursolicacid was dissolved in N,N-Dimethylformamide (DMF) (70 mL). Then, 1.6 g K_2_CO_3_ (11.5 mmol) and 1.4 mL CH_3_I (28 mmol) were added successively.The mixture was stirred for 24 h at room temperature. Then the mixture was poured into water and the resulting suspension was filtrated to give **UA-1** (solid, 3.9 g, 77%). ESI-MS (+) m/z: 493.3 [M + Na]^+^. ^1^H NMR (CDCl_3_, 600 MHz): δ 5.24 (t, 1H, *J* = 6.8 Hz), 3.60 (s, 3H), 3.20 (dd, 1H, *J*_1_ = 20.6 Hz, *J*_2_ = 11.0 Hz), 1.07, 0.98, 0.95, 0.91, 0.85, 0.77, 0.73, (7×s, 7×3H, 23, 24, 25, 26, 27, 29, 30-CH_3_). ^13^C NMR (CDCl_3_, 150 MHz): δ 178.05 (28-C), 138.10 (13-C), 125.53 (12-C), 77.42 (3-C), 55.17, 52.84, 51.43, 48.05, 47.52, 41.96, 39.44, 39.01, 38.83, 38.72, 38.57, 36.93, 36.61, 32.93, 30.62, 28.11, 27.99, 27.19, 24.19, 23.58, 23.27, 21.17, 18.28, 17.01, 16.88, 15.59, 15.41.

*Compound **UA-2***: To the solution of 220 mg (2.2 mmol) of succinic anhydrate and 4-dimethylaminopyridine (DMAP) (15 mg) in 50 mL of pyridine (Pyr), 1.0 g (2.12 mmol) of **UA-1** was added. The reaction solutionwas stirred at 80 °C for 20 h. Then, the mixture was cooled to room temperature and poured in to 100 mL of water. After that, the solution was extracted with dichloromethane and the organic phase was washed with water and brine. It was then dried by MgSO_4_ and the solid was filtered out. The obtained liquid were evaporated and the solids were purified by column chromatography (CH_2_Cl_2_:CH_3_OH = 50:1) forming **UA-2** as a yellow solid (1.05 g, 87%). ESI-MS (–) m/z: 570.7 [M-H]^−1^. ^1^H NMR (CDCl_3_, 600 MHz): δ 5.23 (t, 1H, *J* = 6.2 Hz), 4.52(t, 1H, *J* = 14.4 Hz), 3.61 (s, 3H), 1.31, 1.06, 0.93, 0.86, 0.83, 0.83, 0.78 (7×s, 7×3H×2, 23, 24, 25, 26, 27, 28, 30-CH_3_). ^13^C NMR (CDCl_3_, 200 MHz): δ178.16, 177.09, 171.79, 138.10, 125.40, 81.48, 55.24, 52.79, 51.43, 48.03, 47.39, 41.91, 39.42, 38.97, 38.81, 38.17, 37.66, 36.78, 36.57, 32.82, 30.58, 29.26, 29.02, 28.84, 28.58, 27.94, 24.15, 23.40, 23.24, 21.14, 18.13, 17.02, 16.83, 16.69, 15.42.

*Compound **UA-3***: 1-(3-Dimethylaminopropyl)-3-ethylcarbodiimide Hydrochloride (EDCI) (1.03 mmol) and Boc protected *p*-phenylenediamine (215 mg) were added to the solution of 500 mg (0.88 mmol) of **UA-2** in 8 mL of dry CHCl_3_. Then the reaction mixture was stirred for 32 h at room temperature. After that the mixture was washed with water (20 mL) and brine (20 mL) and dried by MgSO_4_. The solid was filtered out and the obtained liquid was evaporated. The solid was purified by column chromatography (PE:CH_3_CO_2_C_2_H_5_ = 4:1) forming **UA-3** as a yellow solid (315 mg, 47%). ESI-MS (+) m/z: 784.0 [M + Na]^+^. ^1^H NMR (600 MHz, CDCl_3_): δ 7.75 (s, 1H), 7.37 (d, 2H, *J* = 11.0 Hz), 7.27 (d, 2H, *J* = 11.0 Hz), 6.54 (s, 1H),5.23 (s, 1H), 4.51 (m, 1H), 3.59 (s, 3H), 1.48 (s, 9H), 1.21, 0.90, 0.88, 0.87, 0.82, 0.82, 0.69, (7×s, 7×3H, 23, 24, 25, 26, 27, 29, 30-CH_3_). ^13^C NMR (100 MHz, CDCl_3_) δ: 178.03, 172.84, 169.65, 152.86, 138.04, 134.48, 133.25, 125.36, 120.56, 119.15, 81.46, 81.21, 60.32, 55.19, 52.75, 51.37, 47.97, 47.35, 41.86, 39.38, 38.92, 38.76, 37.64, 36.73, 28.27, 28.00, 23.48, 21.08, 16.97, 16.78, 16.66, 15.36.

*Compound **UA-4***: At 0 °C, 2.4 mL of trifluoroacetic acid (TFA) was added to the solution of 500 mg (0.66 mmol) of **UA-3** in 8 mL of CH_2_Cl_2_. The reaction mixture was stirred for 8 h at room temperature. Then the solution was washed with saturated NaHCO_3_, water (30 mL) and brine (30 mL), then dried by MgSO_4_ and evaporated. To obtain a solid, this compound was further purified by column chromatography (CH_2_Cl_2_:CH_3_OH = 100:1), forming **UA-4** as a yellow solid (427 mg, 85%). ESI-MS (+) m/z: 661.5 [M + H]^+^. ^1^H NMR (600 MHz, CDCl_3_): δ 7.64 (s, 1H), 7.23 (s, 2H, *J* = 17.28 Hz), 6.60 (d, 2H, *J* = 17.2 Hz), 5.23 (s, 1H), 4.51 (m, 1H), 3.54 (s, 3H), 1.06, 0.94, 0.91, 0.86, 0.83, 0.83, 0.72 (7×s, 7×3H, 23, 24, 25, 26, 27, 29, 30-CH_3_). ^13^C NMR (200 MHz, CDCl_3_) δ: 178.04, 172.95, 169.50, 142.98, 138.08, 129.32, 125.38, 121.86, 115.34, 81.48, 55.21, 52.77, 51.41, 47.99, 47.38, 41.89, 39.40,38.95, 38.78, 37.67, 36.76, 28.04, 23.72, 22.76, 23.51, 21.12, 17.00, 16.81, 16.70, 15.39.

*Compound **UA-5***: First 200 mg (0.30 mmol) **UA-4** and Et_3_N (55 μL) were dissolved in 12 mL dry tetrahydrofuran (THF) and cooled to 0 °C. Then 35 μL of (0.35 mmol) benzoyl chloride was added to the above solution. The reaction mixture was stirred for 2 h at 50 °C. After that the solution was filtered and evaporated to obtain a solid product. Purification was conductedby silica gel column with CH_2_Cl_2_:CH_3_OH = 40:1 as the eluent to produce **UA-5** as a light yellow solid (195 mg, 85%). HRMS(High-Resolution Mass Spectrum)calculated for C_48_H_65_N_2_O_6_: 765.4843, found: 765.4835. ^1^H NMR (600 MHz, CDCl_3_): δ 8.38, 8.24 (2×s, 2H), 7.86 (d, 2H, *J* = 14.4 Hz), 7.46~7.36 (m, 7H), 5.21 (s, 1H), 4.51 (m, 1H), 3.59(s, 3H), 1.05, 0.93, 0.90, 0.85, 0.83, 0.83, 0.71 (7×s, 7×3H,23, 24, 25, 26, 27, 29, 30-CH_3_). ^13^C NMR (200 MHz, CDCl_3_): δ 178.09, 172.92, 170.00, 165.97, 138.07, 134.69, 134.56, 133.92, 131.69, 128.58, 127.17, 125.38, 121.53, 120.75, 81.54, 55.21, 52.77, 51.43, 48.01, 47.37, 41.89, 39.40, 38.95, 38.79, 37.67, 36.76, 28.06, 23.53, 23.21, 21.13, 17.01, 16.83, 16.72, 15.41.

## 3. Results and Discussion

The gelation behaviors of **UA-5** in solvents were measured by the method of “stable to the inversion of a tube” [30]. Certain amounts of **UA-5** and solvent were put into a sealed glass bottle and heated. Then, the mixture was treated with ultrasound and cooled to room temperature. Subsequently the bottle was inverted to observe if a gel had formed. The observed gelation behaviors are presented in Table 1. Transparent gel formationsin some aromatic solvents such as chlorobenzene, bromobenzene and *o*-dichlorobenzene were observed (Figure 1) and they were all stable at room temperature for several days.The product was soluble in polar aprotic solvents like dimethylformamide (DMF), tetrahydrofuran (THF) and dimethyl sulfoxide (DMSO), and insoluble in some protonic solvents like methanol and water. In nonpolar solvents such as benzene and petroleum ether, it formed aprecipitate state. Only the solvents with suitable polarity could be gelated by the gelators of **UA-5**. As the literature suggests, gelation formeddue to a balance ofgood solubility and the aggregation of the gelators [31]. Mixed solvents were also tested. **UA-5** was soluble in chloroform, in which itformed a clear solution and turned into a transparent gel at room temperature after the gradual addition of toluene or *p*-xylene under sonication. In addition, the gelation property (T_gel_) indicating the required temperature for gel to break was affected by the volume proportions of the solvents. T_gel_ was tested and plotted as afunction of volume proportions of toluene under the same gelator concentration. Obviously, the T_gel_ illustrating the thermal stability of the gel was the maximum at a volume ratio of 2:1 between toluene with chloroform (Figure 2).The ursolic acid derivatives (except **UA-5**) could not form organogel in the tested solvents, perhaps due to the insufficient hydrogen bonding and π–π stacking.

Transmission electron microscopy (TEM) was employed to observe the morphology of supramolecular gels. As shown in Figure 3a, the xerogel in chlorobenzene presented a typical 3D fiber network structure with fiber diameters of about 100 nm. We should note that the nanofiber diameters in different solvent systems were different. Entangled nanofibers with awidth of about 50 nm creating a closely packed 3D network could also be observed in the xerogel from chloroform and toluene above the minimum gelator concentration (MGC) (Figure 3b). Obviously, as the gel became more stable, it exhibited a thinnerfibrous structure and closer entanglement, which was consistent with the stability in different solvents indicated by the MGC values. A reasonable explanation for this was that the close cross-linking network could immobilize more solvent molecules. Under the MGC, **UA-5** could also self-assemble into fibers in chloroform induced by toluene, butthese were too short to entangle together to form an organogel (Figure 3c).

Fourier-transform infrared spectra (FT-IR) and ^1^H NMR experiments were performed to infer the driving forces of the gel formation [32,33,34]. FT-IR spectra showed that in the gel state, the aliphatic C–H stretching vibration band of ursolic acid skeleton appeared at a lower wavenumber (2997 cm^−1^) compared with that in the solid state (3006 cm^−1^). It illustrated that the increased van der Waals interaction of aggregated gelators drive the gel formation. In addition, compared with the solid state, the NH bending (amide II) band shifted from 1552 cm^−1^ to 1542 cm^−1^ when the gel formed. In the solid state, the peak at 1647 cm^−1^ was assigned to the C=O stretching band (amide I) and benzene skeleton vibration, which split into two peaks at 1644 cm^−1^ and 1675 cm^−1^ in the gel (Figure 4). The above results revealed that hydrogen bonding between amide groups and π–π interaction between benzene rings promoted the formation of organogels. The variable temperature ^1^H NMR of **UA-5** spectra in deuterated chloroform and toluene were performed to confirm the driving forces and the results are shown in Figure 5. As the temperature increased from 25 °C to 45 °C, the gel changed into a solution state gradually, while the amide protons and aromatic protons upfieldshifted gradually, implying that the π–π interaction between the aromatic rings and hydrogen bonding played synergic roles for gel formation [15,17].

To explore the mechanical properties of the gel, rheological measurement was conducted on the gel in *o*-dichlorobenzene, in which self-assembly fibers yield a cross-linked three dimensional network. The storage modulus G′ and the loss modulus G″ which characterized the viscoelastic behavior of the gels were measured as functions of shear stress at a constant frequency of 1.0 Hz at 25 °C. As shown in Figure 6, it exhibited a clear thixotropic property and the beginning G′ was about seven times greater than G″, indicating the dominant elastic characteristic of this gel [35,36]. A sudden decrease in the two values was observed above 86 Pa, indicating the breakup of the gel networks under high shearing force—in other words, an indication of a dominant fluidity characteristic.

## 4. Conclusions

Aromatic rings as functional groups were conjugated to ursolic acid and the derivative could self-assemble into a supramolecular gel based on intermolecular hydrogen bonding, π–π stacking interaction as well as the aggregation property of the ursane-triterpenoid skeleton. Introducing functional groups at the A ring, providing non-covalent interactions, proved to be a feasible method to develop supramolecular self-assembly architecture from biocompatible ursolic acid. Moreover, this study expands the building block for self-assembly from natural product triterpenoids. The authors hope to contribute to the design of more functional molecules based on ursolic acid to form supramolecular self-assemblies in the future.

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
