# Peer review of "The Supramolecular Organogel Formed by Self-Assembly of Ursolic Acid Appended with Aromatic Rings"

_materials, 2019, doi:10.3390/ma12040614_

Round 1
Reviewer 1 Report
The manuscript “The supramolecular organogel formed by self-assembly of ursolic acid appended with aromatic rings” by Cui and coworkers described an efficient formation of hydrogel by self-assembly of ursolic acid derivative. This manuscript has covered all the required references. Moreover, the current manuscript clearly demonstrated synthesis of the ursolic acid derivatives. Furthermore, all the experimental details and analytical data are constructed nicely. In addition, the formation of organogel was methodically supported by several experiments, which also explained the properties of the organogel. Therefore, I believe this current manuscript will be useful for material chemistry research. Accordingly, this manuscript is deserved to publish in “Materials” after the following minor revisions. 1. It is suggested to revise the synthetic scheme 1. Please, revise the incorrect structure of UA. Please, provide yield of the reactions in each step. Please, provide reaction conditions in more details. 2. It is shown that ursolic acid derivative UA-5 could form the organogel. Have you look into other ursolic acid derivatives that can potentially form the organogel? How would be their properties? It is suggested to address this matter. Please, include your comment in the manuscript
Author Response
Response to Reviewer 1 Comments Point 1:It is suggested to revise the synthetic scheme 1. Please, revise the incorrect structure of UA. Please, provide yield of the reactions in each step. Please, provide reaction conditions in more details. Response 1:The structure of UA was revised in Scheme 1. The yield of the reactions in each step was added in Scheme 1. The reaction conditions containing reagents, solvents, temperature were presented in Scheme 1. Point 2: It is shown that ursolic acid derivative UA-5 could form the organogel. Have you look into other ursolic acid derivatives that can potentially form the organogel? How would be their properties? It is suggested to address this matter. Please, include your comment in the manuscript . Response 2: The ursolic acid derivatives prepared in this work except UA-5 could not form organogel in the tested solvents due to the insufficient hydrogen bonding and π–π stacking. This comment was added in the manuscript.
Reviewer 2 Report
The manuscript describes the synthesis of ursolic acid (UA) derivatives and their gel formation in uncommon solvents. The paper reports routine experiments, no novelties are added.
1) Scheme 1: Is there any difference between the molecular structure of UA and UA-1 that was obtained in the reaction of UA with CH3I?
2)Several abbreviations are not explain, for example: DMAP, TFA, Pyr; EDCl, EDCJ, HRMS, ....
3)Table 1: The MGC of UA-5 in Br-benzene results higher than in Cl-benzene. Could the solvent properties, (i.e. dipole moment, HB-donor ability or HB-acceptor, dielectric constant, ...) directly related with driven forces between UA-5 and solvent, explain the results? In that case, the solvent properties should appear in Table 1.
4)Figure 3: The comparison of TEM images obtained in the three different solvents cannot be truly compared because the gel concentration (UA-5) is different. Why did not the authors use the same amount?
5)p. 7, line 174: Figure 5d is cited but it is not included ... and Figures 5a, 5b and 5c?
6)The last sentence of conclusions section has not been tested, it should be deleted.
Author Response
Response to Reviewer 2 Comments
Point 1: Scheme 1: Is there any difference between the molecular structure of UA and UA-1 that was obtained in the reaction of UA with CH3I?
Response 1:The structure of UA was incorrect in Scheme 1 and was revised.
Point 2: Several abbreviations are not explain, for example: DMAP, TFA, Pyr; EDCl, EDCJ, HRMS.
Response 2: The abbreviations mentioned have been explained in the manuscript.
Point 3: Table 1: The MGC of UA-5 in Br-benzene results higher than in Cl-benzene. Could the solvent properties, (i.e. dipole moment, HB-donor ability or HB-acceptor, dielectric constant.) directly related with driven forces between UA-5 and solvent, explain the results? In that case, the solvent properties should appear in Table 1.
Response 3: Solvents have effects on the gelation performance of gelators. Because of the stronger electronegativity of atom Cl than Br, the polarity of chlorobenzene is stronger than bromobenzene, which result in better solubility of UA-5 in chlorobenzene. As reported in the literature [Soft Matter, 2013, 9, 7780-7786.], the gelator–solvent interaction was neither too strong nor too weak, which was beneficial to gel formation. But the detail theory was not studied in this paper. We just speculated that only the solvents with suitable polarity could be gelated by UA-5. It confirmed that the gelation occurred as a result of a balance between a tendency of being soluble and an aggregation tendency of the gelators.
Point 4: Figure 3: The comparison of TEM images obtained in the three different solvents cannot be truly compared because the gel concentration (UA-5) is different. Why did not the authors use the same amount?
Response 4: The minimum gelator concentration (MGC) in different solvents was not the same and we should investigate the morphology of the organogel above the MGC. Obviously more stable the gel was, more thin fibrous structure and close entanglement it had, which was consistent with the stability in different solvents indicated by the MGC values. In addition, the morphology was compared between organogels with the same solvent but different concentrations. Under the MGC, UA-5 could also self-assemble into fibers in chloroform induced by toluene, but too short to entangle together forming organogel.
Point 5: p. 7, line 174: Figure 5d is cited but it is not included ... and Figures 5a, 5b and 5c?
Response 5: It should refer to Figure 6 and was revised in the manuscript.
Point 6: The last sentence of conclusions section has not been tested, it should be deleted.
Response 6: It has been deleted.
Reviewer 3 Report
The manuscript deals with the preparation and physico-chemical characterization of ursolic acid and the derivative supramolecular gel based on intermolecular hydrogen bonding, π-π stacking interaction as well as aggregation property of ursane-triterpenoid skeleton.
The manuscript responds to the objectives defined in the title and developed in the introduction, the results are in accordance with the advanced explanation in the discussion part with regards to prepare and to characterize the corresponding organogels.
After a careful proof-reading of the manuscript, this research article is recommended for publication in Matrials with some remarks and suggestions.
- Please specify the reaction condition of each step of UA-5 synthesis.
- For the MGC, could you used “wt %” instead “g/100 cm3”?
- To obtain UA organogels, is it possible to use more eco-friendly solvents?
- Could you give some information about the use control (not jellified system)?
- Is it possible to determine the organogel porosity by BET (Brunauer, Emmett et Teller) theory?
- Rheological experiments in dynamic mode could be complementary to determine the mechanic properties of gels and to confirm the organogel phase transition parameters, as describe in:
Kirilov, P.; Palomo, M.C. Colloidal dispersions of gelled nanoparticles (GLN): concept and potential applications, Gels 2017, 3(3), 33 – 46.
- The statistical analysis is not provided. Please be more precise, show results and discuss them.
- Please describe precisely each technique used (no information about the rheology).
- Please be more precise about the reproducibility and repeatability of each technique used.
- Please include the sources of all these chemicals.
- Organogel physical stability study could be provided.
- FT-IR study could be did according to the temperature to confirm the gelled state.
- 1H RMN results is not clearly explained. Please reformulate.
- References should be checked as some references are not properly formatted.
Author Response
Response to Reviewer 3 Comments
Point 1: Please specify the reaction condition of each step of UA-5 synthesis.
Response 1:The reaction conditions containing reagents, solvents, temperature were presented in Scheme 1.
Point 2:For the MGC, could you used “wt %” instead “g/100 cm3”?
Response 2: Generally, in hydrogel system MGC was expressed by “wt %”[Angew. Chem. Int. Ed. 2015, 54, 5408–5412]. In this paper, the UA-5 could form gel in organic solvents and it was more convenient to use “g/100 cm3” when preparing the gel. In addition, the related references referred to “g/100 cm3” too [New J. Chem., 2014, 38, 6050; RSC Adv., 2013, 3, 23548.]
Point 3: To obtain UA organogels, is it possible to use more eco-friendly solvents? Could you give some information about the use control (not jellified system)?
Response 3: We used the common solvents to test the gel formation and found that it was soluble in polar aprotic solvents like dimethylformamide (DMF), tetrahydrofuran (THF), and dimethyl sulfoxide (DMSO), and insoluble in some protonic solvents like methanol, water. In nonpolar solvents such as benzene and petroleum ether, it was precipitate.
Point 4: Is it possible to determine the organogel porosity by BET (Brunauer, Emmett et Teller) theory?
Response 4: A typical 3D fiber network structure with fiber diameter within 100 nm of the organogel was observed using TEM and the fiber entangled together to produce porosity which could trap solvent molecules forming gel. We could observe the structure by TEM in this paper and if we further study the functions of the xerogel such as adsorption [RSC Adv., 2013, 3, 23548], it is necessary to determine the porosity by BET theory.
Point 5: Rheological experiments in dynamic mode could be complementary to determine the mechanic properties of gels and to confirm the organogel phase transition parameters, as describe in: Kirilov, P.; Palomo, M.C. Colloidal dispersions of gelled nanoparticles (GLN): concept and potential applications, Gels 2017, 3(3), 33–46.
Response 5: As mentioned in this reference “gel–sol phase transition parameters of organogels are measured by following the variation of the elastic modulus (G’) and viscous modulus (G”) at increasing temperatures, which could ensure the gelation process. The literature was added as [35] in the manuscript. In this paper, rheological measurement was used to simply characterize the thixotropic property of the organogel and indicate the dominant elastic character of the gel.
Point 6:The statistical analysis is not provided. Please be more precise, show results and discuss them.
Response 6:According to the gel preparing procedures and the amount errors, the MGC errors were given in the manuscript.
Point 7:Please describe precisely each technique used (no information about the rheology).
Response 7:Rheological measurement was described in the manuscript. Rheological properties were measured by Physica MCR301 rotary rheometer at room temperature.
Point 8: Please be more precise about the reproducibility and repeatability of each technique used.
Response 8: There was no doubt about the synthesis route for UA-5 because each compound was confirmed by NMR and mass spectrum (MS). For other techniques used we did at least three times such as Tgel and MGC tests and the error information was added in the manuscript.
Point 9: Please include the sources of all these chemicals.
Response 9: The sources of all these chemicals were added in the manuscript. Triethylamine (Et3N), benzoyl chloride, toluene, chlorobenzene, o-dichlorobenzene, bromobenzene, DMF were purchased from Beijing Chemical Plant (analytical purity). All solid samples requiring drying were vacuum-dried for 5-10 hours before use.
P-phenylenediamine, 1-ethyl-3-(dimethylaminopropyl) carbonyl diimide hydrochloride (EDCI), trifluoroacetic acid (TFA), isophthaloyl dichloride was purchased from J&K Scientific Co., Ltd.
Point 10: Organogel physical stability study could be provided.
Response 10: Organogels were all stable at room temperature for several days.
Point 11: FT-IR study could be did according to the temperature to confirm the gelled state.
Response 11: FT-IR experiment was carried out under the condition of oxrogel and compared with the state of solid.
Point 12: 1H RMN results is not clearly explained. Please reformulate.
Round 2
Reviewer 2 Report
The revised manuscript has been improved. This reviewer recommends its publication in the present form.
Reviewer 3 Report
The corrections were took in account by the authors. I accept the manuscript to be published in Materials.